# Accumulation of Toxic Elements in Bone and Bone Marrow of Deer Living in Various Ecosystems. A Case Study of Farmed and Wild-Living Deer

**DOI:** 10.3390/ani10112151

**Published:** 2020-11-19

**Authors:** Katarzyna Tajchman, Aleksandra Ukalska-Jaruga, Marek Bogdaszewski, Monika Pecio, Katarzyna Dziki-Michalska

**Affiliations:** 1Department of Animal Ethology and Wildlife Management, Faculty of Animal Sciences and Bioeconomy, University of Life Sciences in Lublin, Akademicka 13, 20-950 Lublin, Poland; katarzyna.michalska@up.lublin.pl; 2Department of Soil Science Erosion and Land Protection, Institute of Soil Science and Plant Cultivation, State Research Institute, Czartoryskich 8, 24-100 Puławy, Poland; aukalska@iung.pulawy.pl (A.U.-J.); mpecio@iung.pulawy.pl (M.P.); 3Institute of Parasitology of the Polish Academy of Sciences, Research Station in Kosewo Górne, 11-700 Mrągowo, Poland; kosewopan@kosewopan.pl

**Keywords:** *Cervus elaphus*, toxic elements, bone, bone marrow, heavy metals, absorption

## Abstract

**Simple Summary:**

Toxic elements (TE) such as Be—beryllium, Al—aluminum, As—arsenic, Cd—cadmium, Sb—antimony, Ba—barium, Pb—lead, V—vanadium, Ni—nickel, Tl—thallium may negatively impact bone cells even at low concentrations. This is especially undesirable when they are released from the bone marrow. Therefore, in this study, the concentrations of TE in the bone marrow and bones of wild and farmed red deer were compared to evaluate the influence of the external environment on the absorption and accumulation of various harmful elements. The obtained results show that higher accumulation was detected only in the case of As, Ba, and Pb in the bones of the wild red deer, compared to the farmed group. In turn, higher levels of Al in the bone marrow and bones, as well as Cd in the bones were recorded in the farmed animals. Although the study involved animals living in an area that is regarded as being unpolluted, the concentrations of some heavy metals were higher than values reported from industrial regions.

**Abstract:**

The aim of the study was to determine the concentrations of toxic elements accumulated in the bone marrow and bones (*Cervus elaphus*). The studies were carried out on two groups of young stags: farmed (*n* = 6) and wild (*n* = 9). Their body weights were measured and bone and bone marrow samples were collected. The concentrations of toxic elements were analyzed using the inductively coupled plasma mass spectrometry technique. The mean aluminum content in the bone marrow and bones of the farmed animals was significantly higher than in the wild group (*p* < 0.05). The mean concentration of arsenic, barium and lead in the bones of the wild red deer was significantly higher than in the bones of the farmed animals (*p* < 0.05), while the cadmium concentration in the bones of the farmed red deer exceeded the value determined in the wild animals. A significant difference was found between the mean concentrations of aluminum, arsenic, barium, lead, vanadium and nickel in the bone marrow and bones of both red deer groups (*p* < 0.05). Although the study involved animals living in an uncontaminated area, the concentrations of some heavy metals were higher than values reported from industrial regions.

## 1. Introduction

Toxic elements (TE), e.g., Be (beryllium), Al (aluminum), As (arsenic), Cd (cadmium), Sb (antimony), Ba (barium), Pb (lead), V (vanadium), Ni (nickel) and Tl (thallium) are transferred from the abiotic environment to living organisms and accumulate in biota at different trophic levels, thereby contaminating food chains/webs. Trophic transfer, bioaccumulation, and biomagnification of TE in food chains have important implications on wildlife and human health. TE can enter the food chain and bioaccumulate in the hard and soft tissues or organs of animals and have a long half-life time in the organism. Moreover, they exhibit a high potential to exert carcinogenic and teratogenic effects and cause renal and immune disorders [1,2]; hence, investigations of their bioaccumulation and biomagnification potential are very important. Generally, metals and metalloids are characterized by chemical persistence and resistance to metabolic transformation leading to elimination thereof from the organism. Although their chemical forms may change as they pass through the intestinal tract or during storage in animal tissues, they exhibit a high sorption affinity to living cells [3]. Therefore, they can pose a high environmental risk for hosts in the trophic chain.

Studies of the feeding habits of wild cervids have shown that leaves, buds, fruit, and flowers of woody plants and shrubs dominate their annual diet. With a few exceptions, grasses represent the least important forage class; however, they are more important in areas with no shrubs and low diversity of woody plants [4,5]. As reported by Janiszewski and Szczepański [6,7], shoots and herbaceous plants constitute, on average 48.68–86.05% and 13.93–51.31%, of the diet of wild deer, respectively. Cervids prefer shrubs and herbaceous vegetation, but consume a low proportion of grasses. Grasses may account for 40% of their diet in autumn and winter but only 5–20% per year [8] depending on the precipitation rates [9]. In contrast, grasses and herbaceous plants are an important component of the diet of farmed deer and are usually consumed at a constant level [10]. The nutritional quality and concentrations of minerals in feed available in the environment may change without a predictable trend depending on the phenological phase of plants and between vegetation classes, types, and density structure [11]. It has been shown that the high variability in the concentrations of mineral compounds in plants depends on various components of plant cell walls, types of soil, precipitation rates, and weather conditions [11,12]. Wild cervids, unlike farmed representatives of this group of animals, are able to move freely and consume forest and cultivated plant species and can thus compensate for deficiencies in the intake of mineral compounds. However, they are largely exposed to compounds accumulated by plants treated with fertilizers and plant protection products or exhaust gases deposited on roadside vegetation [13].

There are many reports on chronic cadmium, zinc, and lead poisoning in domestic and laboratory animals. High intakes of cadmium can cause anemia, enteropathy, and kidney damage. Signs of zinc toxicity include anemia, poor bone mineralization, and arthritis. Lead toxicity is characterized by abnormalities in hematological, neural, renal, or skeletal systems [14,15,16,17]. Very little is known of the effects of these metals on wild mammals or the movement of these metals through wildlife food chains [18]. Large herbivores, e.g., roe deer (*Capreolus capreolus*), red deer (*Cervus elaphus*), reindeer (*Rangifer tarandus*), and moose (*Alces alces*) have been shown to be good bioindicator species in monitoring programs, which have been developed since 1980 in different countries in Central Europe and Scandinavia [19]. Furthermore, it has been shown that deer bone tissues can serve as environmental bioindicators [20,21], especially bones and teeth, which unlike antlers are not replaced every year and accumulate trace elements over years or decades [22]. In humans, for instance, the biological half-life of trace elements in bone tissue lasts up to 30 years, and their content in bones is up to 90% [23]. Moreover, such heavy metals as lead or cadmium are mainly deposited in bone tissues through their interactions with calcium [24]. Bone tissues and the entire organism are built of cells formed in the bone marrow, especially in newborns and young animals, whose bone marrow consists largely of hematopoietically active tissue, with relatively little fat. The bone marrow is the primary site of blood cell production, i.e., hematopoiesis. Hematopoietic tissue is highly proliferative. Billions of cells per kilogram of body weight are produced each day. The hematopoietic system is under exquisite local and systemic control and responds rapidly and predictably to various stimuli [25]. Therefore, any substance present in the bone marrow or such a process as the accumulation of TE can affect the entire organism. Hence, the aim of this study was to determine the concentration of selected toxic elements (Be, Al, As, Cd, Sb, Ba, Pb, V, Ni, Tl) accumulated in the bone marrow and bones of red deer (*Cervus elaphus*), as well as to compare the concentrations of these metals in tissues of wild and farmed red deer.

## 2. Materials and Methods

### 2.1. Experimental Design

The studies were carried out on two groups of red deer: farmed (*n* = 6) and wild (*n* = 9). The analysis included 15 male deer in the first year of life, i.e., 6–7 months year old, as young stags are usually culled in the breeding process, while hinds are intended for reconstruction of the herd. The first group of animals was bred at the Research Station of the Institute of Parasitology, Polish Academy of Sciences, Kosewo Górne (Region of Warmia and Mazury; Poland; N: 53°48′; E: 21°23′). The breeding system was based on rotational pasture within plots with an area and density recommended by DEFRA [26], FEDFA [27], and Mattiello [10]. During the summer grazing period (April to October), the studied animals fed on the available pasture vegetation.

The second group of animals comprised wild individuals living in the area of Strzałowo Forest District located near the deer farm. The forests in the forest district are dominated by coniferous species occupying 89%, whereas deciduous plants cover only 11% of the area. The deer farm and the neighboring forest district are located in an area with the lowest population density, clean air, and 30.9% of forest cover. The study area is characterized by an average annual precipitation in the range of 500–634 mm, average temperature of 7.0–7.7 °C, and a 190–200-day long vegetation season [28]. Moreover, the region of Warmia and Mazury, situated in the northeastern part of the country, is considered an uncontaminated zone, as it is located far from large industrial and urban centers in Poland [29].

### 2.2. Sampling

The body weight of the farmed animals was measured before slaughter as in Tajchman et al. [30]. The weight of the wild red deer was estimated from the carcass weight, which accounts for 67% of the body weight of these animals [31,32,33]. The carcass weight was determined after culling and evisceration of the animals.

Bones and bone marrow were sampled in November 2019. The samples from the farmed animals were collected during routine slaughter, which is the final stage of breeding. The wild red deer individuals were harvested during hunting seasons in accordance with the applicable Rules of Individual and Population Game Animal Selection in Poland (Polish Journal of Laws, Annex to Resolution No. 57/2005 of 22 February 2005).

On the day of slaughter, metatarsal bone (ossa metatarsalia) samples were collected from each animal. The metatarsal bone was dissected by separating the skin, muscles, and tendons with a stainless steel knife. Fresh bones were opened and bone marrow was collected and frozen. The whole bones were dried and comminuted with a dental titanium drills.

### 2.3. Analysis of Mineralstoxic Elements Content in Bone and Bone Marrow

The metal concentrations in the tissues of the red deer were determined with the aqua regia digestion method (0.5 g of freeze-dried sample of tissue ground with a mortar grinder) with the use of the middle pressure (32 bars) microwave digestion system—Mars Xpress from CEM Corp., Matthews, NC, USA. The extracts were analyzed using the inductively coupled plasma mass spectrometry technique (Agilent quadrupole 7500CE ICP-MS equipped with a torch, a micro-mist nebulizer, nickel sampler and skimmer cones, and a double-pass spray chamber). Argon was used as a carrier gas, and hydrogen and helium were used as reaction gases for elimination of interferences. To minimize the matrix effect and ensure long-term stability, all determinations were made in the presence of an internal standard consisting of 1 mg L^−1^ of ^45^Sc, ^89^Y, and ^159^Tb. A blank sample and certified reference material (CRM028-050) were included in the analyses for quality control. The recovery of the analyzed trace elements ranged from 90 to 97%, while the precision of the method defined as a relative standard deviation (RSD) was <3%. The limit of detection (LOD) values were at the level from 0.007 mg kg^−1^ to 0.099 mg kg^−1^.

### 2.4. Statistical Analysis

The results were expressed as the mean value and standard deviation for the repetitions (*n* = 3). The distribution of the studied variables was assessed using the Shapiro–Wilk test. The concentrations of TE in the bones and bone marrow were compared between the groups of the wild and farmed individuals using the Student’s *t*-test (for variables with a normal distribution) and the Mann–Whitney U test (for variables without a normal distribution). The correlations between the deer body weight and TE concentration were evaluated using the r-Pearson correlation coefficient and the Spearman rank correlation coefficient, while the correlations of the TE concentrations in bones and bone marrow were performed using the Student’s *t*-test (for dependent variables) and the Wilcoxon test (for independent variables). All relationships were assessed based on the significance level at *p* < 0.05. The database was compiled and statistical analyses were carried out using Statistica 9.1 software (StatSoft, Poland).

## 3. Results

The mean concentrations of the analyzed TE in the bone marrow and bones were compared between the wild and farmed animals (Table 1). The levels of Be, Cd, Sb, V, and Tl in the bone marrow of the red deer were below the limit of detection, likewise, as were the level of Be in the bones in the wild animals and Tl in the bones of both red deer groups. The mean concentration of Al in the bone marrow and bones in the farmed animals was significantly higher than in the wild individuals (*p* < 0.05), whereas the mean concentration of As in the bones was significantly higher in the wild red deer than in the farmed animals (*p* < 0.05). The mean concentration of Cd in the bone samples differed significantly (*p* < 0.05) and was higher in the farmed red deer than in the other group. An opposite result was found in the case of the Ba and Pb levels in the bone samples. The mean concentrations of these metals were significantly higher in the wild animals than in the farmed red deer (*p* < 0.05). There were no statistically significant differences in the content of As, Ba, Pb, and Ni in the bone marrow and in the level of Be, Sb, V, and Ni in the bones between the red deer groups. Similarly, the differences in the animal body weight were not significant (Table 1).

The mean concentrations of the selected TE in the bone marrow and bones in the wild and farmed red deer were compared (Table 2). There was a significant difference in the mean concentrations of Al, As, Ba, Pb, V, and Ni between the bone marrow and bones of the wild and farmed red deer and in the entire group of the animals (*p* < 0.05). The accumulation of these metals was significantly lower in the bone marrow than in the bone samples (*p* < 0.05). The Cd concentration in the bone marrow and bone tissue differed and was statistically significantly only in the samples from the farmed red deer and in the total samples from all animals (*p* < 0.05). No significant differences were found in the level of Sb (Table 2).

The concentrations of the analyzed TE were compared with the body weight of the animals (Table 3). The higher body weight of animals was accompanied by significantly lower mean levels of Ni in the bone marrow and V in the bones of the wild red deer (*p* < 0.05). In turn, the mean concentration of Sb and Ni in the bones of all animals showed a significant positive correlation with the body weight (*p* < 0.05) (Table 3).

## 4. Discussion

There are not many investigations of the content of TE in the bone marrow of cervids. Significantly higher concentrations of Cd, Pb, Ni, and V and a lower level of As in the bone marrow of wild and farmed deer were only shown in a population of semi-domesticated reindeer from two northern Norwegian counties (Finnmark and Nordland) [34]. Such analyses of bone marrow in reindeer were carried out, as it is a delicacy for many breeders of this species of cervids. Bone marrow plays a very important role in organisms, since its cells are involved in the remodeling of, e.g., the skeletal system during hematopoiesis [35]. Bone tissue, which is mainly composed of carbonated hydroxyapatite, is remodeled throughout the animal’s life [36], and microelements contained therein can be transported to other tissues or excreted from the organism and new elements from the bone marrow can be incorporated. Bone tissue can therefore serve as a reservoir of metals, as those accumulated in this tissue are released into the bloodstream during the remodeling process. There are little exact quantitative data on the rate of bone system remodeling in cervids, most of them relate to the movement of minerals and other substances from bone into the antler as it grows [36]. In humans, it has been estimated that over 10% of all bone tissue is replaced each year, and the entire skeleton is remodeled within less than 10 years [35]. In animals with a shorter life span, such as cervids, the period of complete skeleton remodeling may be even shorter. This was confirmed by the study on the red deer, which showed a very low concentration of Al, As, Ba, Pb, V, and Ni in the bone marrow, in comparison with bones.

Bone tissue requires a longer remodeling process than soft tissues and can thus be a reservoir of trace elements in the conditions of long-term exposure to environmental pollution [22,23,37]. The concentration of As in the bones of the studied red deer was lower than in investigations of small mammals (with a control group) living near a refinery in Merseyside, England [38]. However, the concentration of this metal in the bones was higher in the samples from the wild deer and lower in the farmed deer in comparison with levels determined in samples from caribou living near a mine in Northern Alaska [39]. The bone Ba concentration was higher in the wild red deer and lower in the farmed animal group than the levels determined in roe deer from forest and field habitats of central Europe [20,40]. The concentrations of Cd and Pb in the bones of reindeer living in Karelia (Russia) were much higher than those reported in the present study [13]. In turn, the concentration of Pb in bones of young deer living in a mining area contaminated with this metal was lower than in the wild and farmed animals in the present study. Moreover, even adult individuals from the contaminated areas had lower Pb levels than the young wild-living deer from Poland [41]. This is surprising, as the animals examined in the present study lived in an unpolluted area. However, as shown by Falandysz et al. [29], the content of Cd in the kidneys of wild deer from the regions of Warmia and Mazury, exceeded the tolerance limits for consumers recommended in Poland. Additionally, the concentration of lead in meat from many carcasses exceeded the allowable limit, which was probably caused by contamination with fine lead particles from bullets [29]. In turn, in comparison with the present results, substantially higher Cd and Pb levels in the bones of young deer were shown in a study conducted in the Netherlands [19]. Pb concentrations determined in the bones of animals living in the mining district in Oklahoma were lower than the values in the wild red deer and similar to the content of this metal in the farmed group [42]. The concentration of Pb in the bones of the wild red deer was higher than that in roe deer from field and forest habitats in central Europe. In turn, the Pb content in the farmed animals was lower than in the roe deer from the forest habitats and similar to the level determined in the roe deer living in the fields [20,40]. The content of Cd and Pb in the red deer bones was lower in comparison with the results obtained in the caribou from Northern Alaska living near the mine [39].

Some metals even at very low concentrations exert an anabolic effect on bone tissue, similar to nickel. They mainly serve as cofactors of enzymes involved in bone remodeling processes. However, both the absence and excess of these metals in the organism can disrupt bone integrity. Whether their effect is beneficial or toxic depends on external (environmental concentration and nutrition) and internal (absorption and metabolism of elements, genetic predisposition, age, and sex) factors and their mutual interactions [43]. Other metals, e.g., cadmium, arsenic, and aluminum, are toxic to bone cells even at low concentrations. Most of the metals analyzed in the present study occur at low concentrations in nature. However, human activity is the major cause of contamination, and exposure to these trace elements causes serious health problems. Bone tissue undergoes continuous life-long remodeling. This process involves a coordinated action of the resorption, synthesis, and mineralization of the bone matrix. In general, metals pose two problems: direct toxicity to bone cells on the one hand and accumulation in the bone matrix on the other. The direct toxicity mainly affects osteoblasts formed in the bone marrow by inhibition of their differentiation, synthesis, and mineralization of the extracellular matrix. Their effects on osteoclasts, i.e., enhancement or reduction in the activity of the tartrate-resistant acid phosphatase enzyme and inhibition of precursor maturation, vary between metals. This leads to an imbalance in the bone remodeling process, reduces bone formation, and contributes to development of bone diseases such as osteopenia and osteoporosis [44]. Moreover, the ability of trace metals to accumulate in the extracellular bone matrix facilitates their bioaccumulation and thus, extends the metal half-life in the organism. This is especially important at continuous exposure to low levels of the metals, as it can produce comparable or worse adverse effects than short exposure to high metal levels [44].

A study conducted by Odstrcil et al. [45] demonstrated that poisoning with low concentrations of arsenic inhibited endochondral ossification in long bones in rats associated with an increase in the width of cartilage, especially in the hypertrophic zone. Moreover, Hu et al. [46] showed that in vivo arsenic trioxide poisoning decreased osteoclast precursor maturation and osteoclast activity, altering the bone resorption process. Additionally, an in vitro study showed arsenic-induced inhibition of proliferation and induction of apoptosis of bone marrow-derived mesenchymal cells [47], osteoblasts, and chondrocytes [48]. Arsenic was also found to increase the generation of reactive oxygen species in osteoblasts [49] and induce apoptosis. This phenomenon may be even more dangerous for cervids, which suffer from cyclic physiological osteoporosis during antler growth [50,51,52,53,54].

The higher content of toxic elements in the tissues of the wild red deer than in the farmed animals was predictable; however, it was detected only in the case of As, Ba, and Pb accumulated in bones. All these metals are of anthropogenic origin. Arsenic enters the natural environment through combustion of fossil fuels. Barium is one of the elements with an undetermined biological role. When used as a rubber and plastic filler, it is probably deposited from car wheels onto plants growing along roads or may be present in water used in various industries. Lead is used in the production of petrol [55] and is thus contained in vehicle exhaust gases deposited on roadside vegetation.

Interestingly, higher levels of Al in the bone marrow and bones and higher contents of Cd in the bones were detected in the farmed red deer. Aluminum is a ubiquitous metal in the Earth’s crust [56]. In turn, the low amount of Cd in the bones of the farmed deer was probably associated with the fertilization of farm pastures [57]. As already mentioned, aluminum is dangerous to the skeletal system. Once absorbed, it is incorporated into the bone matrix and taken up by osteoclasts during the resorption process. It is deposited on trabecular bone surfaces and on the surfaces of vascular canals permeating compact bone. It has also been found on periosteal and endosteal surfaces [58]. In vivo studies have shown that Al deposition in bone decreased Ca, Mg, and P levels, inhibiting the bone mineralization process [59].

As in a study conducted by Grace et al. [60], the average content of TE (Ni in bone marrow and V in bones) in tissues decreased with the increase in the body weight of the red deer, whereas the concentration of Sb and Ni in their bones increased with the higher body weight. This is probably related to the fact that Sb accumulates most easily in bones and lungs [61]. In turn, nickel does not normally accumulate in tissues due to the usually efficient excretion of Ni from the body [62]. The main determinant of Ni toxicity and carcinogenicity is its ability to penetrate into cells and release ions [63].

## 5. Conclusions

TE (Al, As, Ba, Pb, V, and Ni) accumulated in deer bones may originate from bone marrow, given their higher concentration in hard tissues. However, this should be investigated more comprehensively. In the bones of the wild red deer, compared to the farmed group, higher accumulation was detected only in the case of As, Ba, and Pb. In turn, higher levels of Al in the bone marrow and bones, as well as Cd in the bones were recorded in the farmed animals. The concentration of Ni in the bone marrow and V in the bones of the wild red deer declined with increasing body weight, while the concentrations of Sb and Ni in the bones in all animals were proportional to their body weight. Although the investigations were carried out in an area that is regarded as uncontaminated, the concentrations of some toxic elements in the animal tissues were higher and comparable to the areas of high contamination levels.

## Figures and Tables

**Table 1 animals-10-02151-t001:** Comparison of the Toxic elements (TE) concentrations in the bones and bone marrow between the wild and farmed red deer.

Analyzed Parameters			Wild Red Deer		Farm Red Deer		t ^a^/Z ^b^	*p*
			M	SD	M	SD		
Bone marrow	Be	mg/kg	<LOD	<LOD	<LOD	<LOD	-	-
	Al		0.685	0.186	2.638	0.753	4.000 ^b^	0.004 *
	As		0.008	0.002	0.003	0.001	2.093 ^a^	0.056
	Cd		<LOD	<LOD	<LOD	<LOD	-	-
	Sb		<LOD	<LOD	<LOD	<LOD	-	-
	Ba		0.969	0.321	1.077	0.371	−0.215 ^a^	0.833
	Pb		0.003	0.001	0.003	0.003	26.000 ^b^	0.954
	V		<LOD	<LOD	<LOD	<LOD	-	-
	Ni		0.042	0.023	0.014	0.006	16.000 ^b^	0.224
	Tl		<LOD	<LOD	<LOD	<LOD	-	-
Bone	Be		<LOD	<LOD	0.002	0.001	22.500 ^b^	0.607
	Al		2.028	0.350	26.229	4.396	−6.823 ^a^	<0.001 *
	As		0.231	0.014	0.046	0.007	10.171 ^a^	<0.001 *
	Cd		0.001	0.001	0.003	0.0003	6.000 ^b^	0.012 *
	Sb		0.0004	0.0003	0.005	0.002	15.000 ^b^	0.181
	Ba		238.951	38.158	87.978	5.366	3.176 ^a^	0.007 *
	Pb		0.977	0.136	0.545	0.095	2.344 ^a^	0.035 *
	V		0.093	0.027	0.119	0.033	−0.616 ^a^	0.548
	Ni		0.375	0.028	1.667	1.334	26.000 ^b^	0.954
	Tl		<LOD	<LOD	<LOD	<LOD	-	-
Body mass		kg	46.5	2.916	49.5	2.091	16.500 ^b^	0.224

^a^—the Student’s *t*-test result, ^b^—Mann–Whitney test results, M—mean, SD—standard deviation, <LOD—below the limit of detection, * values statistically significant *p* < 0.05.

**Table 2 animals-10-02151-t002:** Comparison of the concentrations of TE between the bone marrow and bones of the wild and farmed red deer.

Comparison of Measurements in Bone Marrow and Bone	Wild Red Deer		Farm Red Deer		All	
	t ^a^/Z ^b^	*p*	t ^a^/Z ^b^	*p*	t ^a^/Z ^b^	*p*
Be	-	-	-	-	-	-
Al	2.310 ^b^	0.021 *	−5.457 ^a^	0.002 *	3.237 ^b^	0.001 *
As	−16.184 ^a^	<0.001 *	−5.314 ^a^	0.003 *	−6.046 ^a^	<0.001 *
Cd	-	-	2.201 ^b^	0.027 *	2.366 ^b^	0.017 *
Sb	-	-	1.603 ^z^	0.108	1.825 ^b^	0.076
Ba	−6.274 ^a^	<0.001 *	−16.703 ^a^	<0.001 *	3.407 ^b^	<0.001 *
Pb	−7.185 ^a^	<0.001 *	−5.817 ^a^	0.002 *	−7.724 ^a^	<0.001 *
V	−3.486 ^a^	0.008 *	−3.657 ^a^	0.015 *	3.407 ^b^	<0.001 *
Ni	2.665 ^b^	0.007 *	2.201 ^b^	0.027 *	3.407 ^b^	<0.001 *
Tl	-	-	-	-	-	-

^a^—the Student’s *t*-test result, ^b^—Wilcoxon pair-order test, M—mean, SD—standard deviation, * values statistically significant *p* < 0.05.

**Table 3 animals-10-02151-t003:** Comparison of the concentrations of TE in the bone marrow and bones with the body weight of the wild and farmed red deer.

Analyzed Parameters		Wild Red Deer Body Mass		Farm Red Deer Body Mass		All	
		R ^a^/r ^b^	*p*	R ^a^/r ^b^	*p*	R ^a^/r ^b^	*p*
Bone marrow	Be	-	-	-	-	-	-
	Al	−0.468 ^a^	0.203	0.257 ^a^	0.623	0.112 ^a^	0.689
	As	0.577 ^a^	0.103	0.179 ^b^	0.734	0.100 ^a^	0.722
	Cd	-	-	-	-	-	-
	Sb	-	-	-	-	-	-
	Ba	0.393 ^a^	0.295	−0.317 ^b^	0.540	0.109 ^a^	0.698
	Pb	0.144 ^a^	0.711	−0.135 ^a^	0.798	−0.016 ^a^	0.953
	V	-	-	-	-	-	-
	Ni	−0.728 ^a^	0.026 *	0.509 ^b^	0.302	−0.296 ^a^	0.282
	Tl	-	-	-	-	-	-
Bone	Be	-	-	0.654 ^a^	0.158	0.371 ^a^	0.172
	Al	0.292 ^a^	0.444	−0.134 ^b^	0.800	0.407 ^a^	0.131
	As	−0.025 ^a^	0.948	−0.170 ^b^	0.747	−0.083 ^b^	0.768
	Cd	0.343 ^a^	0.365	−0.257 ^a^	0.623	0.314 ^a^	0.253
	Sb	0.550 ^a^	0.124	0.394 ^a^	0.438	0.583 ^a^	0.022 *
	Ba	0.561 ^a^	0.116	−0.548 ^b^	0.260	−0.146 ^a^	0.601
	Pb	−0.058 ^a^	0.881	0.462 ^b^	0.356	−0.208 ^b^	0.457
	V	−0.778 ^a^	0.013 *	0.393 ^b^	0.441	−0.316 ^a^	0.250
	Ni	0.518 ^a^	0.152	0.714 ^a^	0.111	0.522 ^a^	0.045 *
	Tl	-	-	-	-	-	-

^a^—Spearman rank-order correlations, ^b^—Pearson r correlations, * statistically significant values at *p* < 0.05.

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
