# Peer review of "Accumulation of Toxic Elements in Bone and Bone Marrow of Deer Living in Various Ecosystems. A Case Study of Farmed and Wild-Living Deer"

_animals, 2020, doi:10.3390/ani10112151_

Round 1

Reviewer 1 Report

This manuscript addresses an important topic that may build on previous research and provide new information to improve scientific knowledge. The title reflects the content of the article.

Summary - needs correction (too general). The summary does not include all the conclusions contained in the article.

The research was conducted using modern and well-selected methods that guarantee the reliability of the obtained results. The statistical methods are appropriate.

Detailed comment:

line 178 in Table 1    ta/Zb     should be explained below the table

line 190 in Table 2     ta/Zb     should be explained below the table

line 191  „b – test kolejnoÅ›ci par Wilcoxona” - must be translated into English

line 199  in Table 3      Ra/rb     should be explained below the table

Author Response

Authors response to manuscript animals-993145 "Accumulation of toxic elements in bone and bone marrow of deer living in various ecosystems. A case study of farmed and wild-living deer"

Authors would like to thank the Reviewer for all valuable comments that increase the quality of this manuscript. All suggestions were included in the revised version of the manuscript. The conclusions have been rewritten to more detailed reflect the results of our research. Moreover, all unexplained abbreviations and characters has been additionally described under the tables.

  1. This manuscript addresses an important topic that may build on previous research and provide new information to improve scientific knowledge. The title reflects the content of the article.                                                      Response: Thank you for your comment. 
  2. Summary - needs correction (too general). The summary does not include all the conclusions contained in the article.                                                  Response: It was improved. Please see lines: 300-309.
  3. The research was conducted using modern and well-selected methods that guarantee the reliability of the obtained results. The statistical methods are appropriate.                                                                               Response: Thank you for your comment. 
  4. Detailed comment:

    line 178 in Table 1    ta/Zb     should be explained below the table  Response: It was improved. Please see line: 174.
  5. line 190 in Table 2     ta/Zb     should be explained below the table Response: It was improved. Please see line: 187.
  6. line 191  „b – test kolejnoÅ›ci par Wilcoxona” - must be translated into English                                                                                       Response: It was improved. Please see line: 187.
  7. line 199  in Table 3      Ra/rb     should be explained below the table Response: It was improved. Please see line: 198.

Reviewer 2 Report

Tajchman, et al Animals

This paper contains a useful set of data, although the reasons for these high concentrations of TE accumulated in bone and marrow in animals grazing well away from industrial centres are not explained.

  1. Introduction: covers the main points but is somewhat verbose. It could be shortened by removing material dealing with diet composition, especially as this is not connected (in the Introduction or Discussion) with reasons for the high TE contents reported here.
  2. Methods: use of a dentist drill to comminute the bones – in every case, the concentrations of minerals in the bone were higher than in the marrow, could this be due to contamination of bone with metal fragments from the drill? This question could possibly be resolved by analysing the composition of the drills used.
  3. Results: straightforward, but Table 2 would be easier to read if the mean elements values were repeated in this table.
  4. Discussion: this is generally well done, although it would be useful if the authors could suggest reasons why the mineral contents in these deer were unexpectedly high. For example, is it possible that prevailing winds moved heavy metals from industrial centres towards the experimental area?

(a) Lines 212-213 and 218-219: “There are no quantitative data on the rate of bone system remodeling in cervids.” The rates of Ca and P inclusion in skeleton and antlers are described by several authors -- see references in Dryden, G.McL. (2016). Animal Production Science, 56, 962-970. http://dx.doi.org/10.1071/AN15051. The relative concentrations of minerals in bone and marrow can’t be taken as to indicate the rate of bone remodelling.

(b) Lines 232-233: “However, as shown by Falandysz et al. [25], the content of Cd in kidneys of deer from the same area exceeded the tolerance limits recommended in Poland.” Please clarify which deer’s kidneys and which area you are talking about, and tolerance levels for what – humans or animals.

  1. Conclusions: line 308: “anthropopressure” – can you find a simpler term than this?

Author Response

Authors response to manuscript animals-993145 "Accumulation of toxic elements in bone and bone marrow of deer living in various ecosystems. A case study of farmed and wild-living deer"

Authors would like to thank the Reviewer for all valuable comments that increase the quality of this manuscript. All suggestions were included in the revised version of the manuscript. The conclusions have been rewritten to more detailed reflect the results of our research. Moreover, all unexplained abbreviations and characters has been additionally described under the tables.

  1. This paper contains a useful set of data, although the reasons for these high concentrations of TE accumulated in bone and marrow in animals grazing well away from industrial centres are not explained.                                Response:  Thank you for the comments. Please see the point 5.
  2. Introduction: covers the main points but is somewhat material dealing with diet composition, especially as this is not connected (in the Introduction or Discussion) with reasons for the high TE contents reported here.verbose. It could be shortened by removing.                                                  Response: The introduction has been shortened as recommended.
  3. Methods: use of a dentist drill to comminute the bones – in every case, the concentrations of minerals in the bone were higher than in the marrow, could this be due to contamination of bone with metal fragments from the drill? This question could possibly be resolved by analysing the composition of the drills used.                                                                          Response: Titanium drills were used for bone fragmentation, in which part of the drill bit face was covered with additional thick layer of titanium nitride. However, this element was not the subject of our research, so we exclude the possibility of TE transfer from the used 'tool' to the bone samples.
    It was added, please see line: 131.
  4. Results: straightforward, but Table 2 would be easier to read if the mean elements values were repeated in this table.                                      Response: All these data are presented in table 1. Supplementing table 2 would involve duplicating the data and unreadable the table 2.
  5. Discussion: this is generally well done, although it would be useful if the authors could suggest reasons why the mineral contents in these deer were unexpectedly high. For example, is it possible that prevailing winds moved heavy metals from industrial centres towards the experimental area? Response: Of course, this may be one of the reasons, but we do not have the results of research on the contamination of the land on pastures used by deer. If there is pollution in this area, it may be related to the so-called low emissions observed in rural areas, e.g. combustion, over-fertilization, excessive use of plant protection products, etc. Nevertheless, one of the additional causes may be the diet and the related feed preservatives containing small amounts of metal residues which are not excreted from the body and accumulate in the bones over the years (especially in the first period of animal growth).
  6. (a) Lines 212-213 and 218-219: “There are no quantitative data on the rate of bone system remodeling in cervids.” The rates of Ca and P inclusion in skeleton and antlers are described by several authors -- see references in Dryden, G.McL. (2016). Animal Production Science, 56, 962-970. http://dx.doi.org/10.1071/AN15051. The relative concentrations of minerals in bone and marrow can’t be taken as to indicate the rate of bone remodelling.                                                                                      Response: It was improved. Please see lines: 204-211.
  7. (b) Lines 232-233: “However, as shown by Falandysz et al. [25], the content of Cd in kidneys of deer from the same area exceeded the tolerance limits recommended in Poland.” Please clarify which deer’s kidneys and which area you are talking about, and tolerance levels for what – humans or animals. Response: It was improved. Please see lines: 230-231.
  8. Conclusions: line 308: “anthropopressure” – can you find a simpler term than this?                                                                                            Response: The word has been interchangeably used by the phrase ‘areas of high contamination levels’.

Reviewer 3 Report

This is a interesting and well-presented study that provides a valuable contribution to the knowledge on the negative influence of human activity over environment, but also gives some insights into the influence of environmental pollution upon the deer skeleton and bone marrow in the context of cervid peculiar physiology (the annually shed antlers in male deer). I found this paper very clear, well organized, the research methods are clearly described and the conclusions are well-founded.

I have just few suggestions that I hope will be useful:

  1. Abstract, lines 30-32: "Their body weight was measured and bone marrow and bone samples." - I don't understand very well this sentence: do you mean that bone marrow and bone samples were extracted?
  2. Introduction, lines 80-83: please, add references here.
  3. the line 91: "the biological half-life of trace elements in bone tissue is even 30 years" - I am not a native English speaker, but this does not sound good for me. May be to replace by "attains 30 years", or "lasts up to 30 years"?

As you can see, the suggestions are very minor, therefore I am happy to recommend this manuscript for publication.

Author Response

Authors response to manuscript animals-993145 "Accumulation of toxic elements in bone and bone marrow of deer living in various ecosystems. A case study of farmed and wild-living deer"

Authors would like to thank the Reviewer for all valuable comments that increase the quality of this manuscript. All suggestions were included in the revised version of the manuscript. The conclusions have been rewritten to more detailed reflect the results of our research. Moreover, all unexplained abbreviations and characters has been additionally described under the tables.

  1. This is a interesting and well-presented study that provides a valuable contribution to the knowledge on the negative influence of human activity over environment, but also gives some insights into the influence of environmental pollution upon the deer skeleton and bone marrow in the context of cervid peculiar physiology (the annually shed antlers in male deer). I found this paper very clear, well organized, the research methods are clearly described and the conclusions are well-founded.            Response: Thank you for your comment. 
  2. Abstract, lines 30-32: "Their body weight was measured and bone marrow and bone samples." - I don't understand very well this sentence: do you mean that bone marrow and bone samples were extracted?           Response: Yes, animals were weighed and bone and bone marrow samples from bones were collected after slaughter.
  3. Introduction, lines 80-83: please, add references here.                  Response: It was added, please see lines: 78, 347-354.
  4. the line 91: "the biological half-life of trace elements in bone tissue is even 30 years" - I am not a native English speaker, but this does not sound good for me. May be to replace by "attains 30 years", or "lasts up to 30 years"? Response: It was improved, please see line: 86.
  5. As you can see, the suggestions are very minor, therefore I am happy to recommend this manuscript for publication.                                    Response: Thank you 

Round 2

Reviewer 2 Report

Thanks for sending the revised paper. I'm happy to recommend that it be accepted for publication.